# Markers of Gut Barrier Function and Microbial Translocation Associate with Lower Gut Microbial Diversity in People with HIV

**DOI:** 10.3390/v13101891

**Published:** 2021-09-22

**Authors:** Ronald J. Ellis, Jennifer E. Iudicello, Robert K. Heaton, Stéphane Isnard, John Lin, Jean-Pierre Routy, Sara Gianella, Martin Hoenigl, Rob Knight

**Affiliations:** 1Departments of Neurosciences and Psychiatry, University of California, San Diego, CA 92093, USA; 2Department of Psychiatry, University of California, San Diego, CA 92093, USA; jiudicello@health.ucsd.edu (J.E.I.); rheaton@health.ucsd.edu (R.K.H.); 3Research Institute of the McGill University Health Centre, Montreal, QC H4A 3J1, Canada; stephane.isnard@mail.mcgill.ca (S.I.); john.lin@mail.mcgill.ca (J.L.); jean-pierre.routy@mcgill.ca (J.-P.R.); 4Department of Medicine, University of California, San Diego, CA 92093, USA; gianella@health.ucsd.edu (S.G.); mhoenigl@health.ucsd.edu (M.H.); 5Department of Pediatrics, University of California, San Diego, CA 92093, USA; rknight@health.ucsd.edu

**Keywords:** HIV, gut microbial diversity, gut barrier dysfunction, microbial translocation, occludin

## Abstract

People with human immunodeficiency virus (HIV) (PWH) have reduced gut barrier integrity (“leaky gut”) that permits diffusion of microbial antigens (microbial translocation) such as lipopolysaccharide (LPS) into the circulation, stimulating inflammation. A potential source of this disturbance, in addition to gut lymphoid tissue CD4+ T-cell depletion, is the interaction between the gut barrier and gut microbes themselves. We evaluated the relationship of gut barrier integrity, as indexed by plasma occludin levels (higher levels corresponding to greater loss of occludin from the gut barrier), to gut microbial diversity. PWH and people without HIV (PWoH) participants were recruited from community sources and provided stool, and 16S rRNA amplicon sequencing was used to characterize the gut microbiome. Microbial diversity was indexed by Faith’s phylogenetic diversity (PD). Participants were 50 PWH and 52 PWoH individuals, mean ± SD age 45.6 ± 14.5 years, 28 (27.5%) women, 50 (49.0%) non-white race/ethnicity. PWH had higher gut microbial diversity (Faith’s PD 14.2 ± 4.06 versus 11.7 ± 3.27; *p* = 0.0007), but occludin levels were not different (1.84 ± 0.311 versus 1.85 ± 0.274; *p* = 0.843). Lower gut microbial diversity was associated with higher plasma occludin levels in PWH (r = −0.251; *p* = 0.0111), but not in PWoH. A multivariable model demonstrated an interaction (*p* = 0.0459) such that the correlation between Faith’s PD and plasma occludin held only for PWH (r = −0.434; *p* = 0.0017), but not for PWoH individuals (r = −0.0227; *p* = 0.873). The pattern was similar for Shannon alpha diversity. Antiretroviral treatment and viral suppression status were not associated with gut microbial diversity (ps > 0.10). Plasma occludin levels were not significantly related to age, sex or ethnicity, nor to current or nadir CD4 or plasma viral load. Higher occludin levels were associated with higher plasma sCD14 and LPS, both markers of microbial translocation. Together, the findings suggest that damage to the gut epithelial barrier is an important mediator of microbial translocation and inflammation in PWH, and that reduced gut microbiome diversity may have an important role.

## 1. Introduction

The gut epithelial barrier is maintained by tight junctions (TJs), cellular border structures that allow the paracellular transport of some solutes and molecules while protecting against toxins in the gut lumen diffusing into the circulation, including lipids and microbial-derived peptides [1,2]. The gut barrier also contributes to the maintenance of symbiotic relationships with commensal gut microbiota. Thus, the gut microbiome and gut barrier integrity have beneficial reciprocal interactions [3,4]. Occludin is one of a number of proteins that make up TJs [5,6,7,8], and expression of occludin is one of the most characteristic structural markers of tight junctions in polarized gut epithelial cells [9]. Pathologic states such as inflammatory bowel disease are associated with a leaky gut epithelial barrier, disruption of TJs, and dysregulation of occludin. In mice, knockdown of occludin using small-interfering RNA transfection caused an increase in transepithelial flux of urea, mannitol, inulin, and dextran [5], indicating that occludin expression is important for the maintenance of the intestinal TJ barrier. A prior study found that observed transcript levels of TJ components including occludin were significantly decreased in the colon in people with human immunodeficiency virus (HIV) (PWH) [10], possibly resulting from TJ damage, with resulting release of occludin into plasma. Additionally, The viral protein Tat decreases production of occludin at endothelial tight junctions [11,12]. HIV remains transcriptionally active even in virally suppressed individuals, producing the Tat protein, though it does not produce infectious virions [13,14]. Furthermore, occludin depletion leads to a preferential increase in the flux of large molecules, including potential toxins of microbial or environmental origin. In people with HIV (PWH), gut barrier integrity is compromised [15,16]. This leads to translocation into the systemic circulation of microbial antigens such as lipopolysaccharide (LPS) and (1→3)-β-D-Glucan [16,17]. These in turn elicit systemic inflammation [17,18,19]. Higher systemic levels of soluble CD14 (sCD14) and LPS are markers of increased microbial translocation and are associated with systemic and CNS inflammation, immune activation [5], and neurocognitive impairment [20].

Alterations in the variety and relative abundance of microbial species, particularly when associated with disease, is termed gut dysbiosis. Gut dysbiosis is often associated with a reduction in microbial diversity. Substantial evidence suggests that metabolites generated by the gut microbiota play a central role in gut barrier integrity [21]. Specific metabolites include short chain fatty acids such as butyrate [22,23,24] and the neurotransmitter tryptophan [25]. Many bacterial species are capable of synthesizing the bacterial enzymes that generate the specific metabolites involved in gut barrier integrity [26]. Thus, a more diverse gut microbiome, not necessarily particular microbial taxa, ensures the presence of the enzymes required to synthesize these beneficial metabolites. Further corroboration of this view comes from the observation that treatment with different antibiotics, which variably affects the relative abundance of different gut microbial species, reduces gut microbial diversity and leads to abnormal gut barrier integrity, rather than the specific bacterial taxa. In light of these considerations, we hypothesized that a reduction in microbial diversity might underlie the known disturbances in gut barrier integrity seen in virally suppressed PWH despite viral suppression and immune recovery [27]. We used plasma occludin levels as an index of abnormal gut barrier integrity, based on the notion that compromise of the gut barrier leads to compensatory overexpression of occludin to facilitate tightening of the barrier [28]. We thus expected reductions in gut microbial diversity to be correlated with higher levels of occludin in blood plasma.

## 2. Materials and Methods

Participants were consecutively enrolled PWH and PWoH research volunteers from community sources at a single site who agreed to provide stool and blood samples and undergo microbiome characterization between May 2015 and May 2019. The MedMira Rapid Test (Halifax, NS, Canada) was used to determine positive or negative HIV and hepatitis C virus (HCV) serostatus. Exclusion criteria were active neurological illnesses other than HIV, such as multiple sclerosis, and active psychiatric or substance use disorder (e.g., psychosis) that might interfere with completing study evaluations. To maximize the representativeness of the cohort, no other exclusions were applied. Informed consent was obtained from all subjects involved in the study (UC San Diego Institutional Review Board #182064).

HIV disease, treatment characteristics and laboratory assays. Clinical and laboratory data were ascertained via comprehensive neuromedical evaluations consisting of a structured clinician-administered interview, physical and neurological examinations, and standard laboratory assays as previously described [29]. Blood was drawn by venipuncture. Participants were not fasted prior to venipuncture. Levels of HIV RNA in plasma were measured via reverse transcriptase-polymerase chain reaction (Amplicor, Roche Diagnostics, Indianapolis, IN, USA) and were considered undetectable below the lower limit of quantitation of 50 copies/mL. Nadir CD4+ T-cells were by self-report and current CD4 by flow cytometry. CD4+ T lymphocyte recovery on ART was calculated as the difference between the current and the nadir CD4. LPS was measured by ELISA (Cusabio; https://www.cusabio.com; accessed on 21 September 2021). Soluble CD14 (sCD14) was quantified using assay kit from R&D (https://www.rndsystems.com/products/human-cd14-quantikine-elisa-kit_dc140, accessed on 21 September 2021). Occludin was measured by ELISA (Cosmo Bio; https://www.cosmobiousa.com/products/human-occludin-ocln-elisa-kit, accessed on 21 September 2021).

Characterization of the gut microbiome. Stool was collected according to a standardized protocol. Participants unable to provide specimen at the on-site visit were provided with supplies (paper commode for specimen collection, fecal collection tube with screwcap and spoon, large ziplock specimen bag, brown paper bag, labels, spill pads, nitrile powder free gloves, ice pack, insulated bag for transport, instructions) to collect and freeze stool off site and return it within 24-h. Stool samples were aliquoted into 5 equal parts, one gram was homogenized and processed in a nucleic acid preservative and stored at -80C for 16S rRNA gene amplicon sequencing. Gut microbial diversity was characterized using 16S rRNA sequencing and indexed by Faith’s phylogenetic diversity (PD) [30,31]; we performed a secondary parallel analysis using the Shannon index of alpha diversity. DNA extraction and 16S rRNA gene amplicon sequencing both were done using Earth Microbiome Project (EMP) standard protocols (http://www.earthmicrobiome.org/protocols-and-standards/16s, accessed on 21 September 2021). DNA was extracted with the Qiagen MagAttract PowerSoil DNA kit (https://www.qiagen.com/us/shop/new-products/magattract-powersoil-dna-isolation-kit/, accessed on 21 September 2021) as previously described [5]. Amplicon PCR was performed on the V4 region of the 16S rRNA gene using the primer pair 515f to 806r with Golay error-correcting barcodes on the reverse primer. Amplicons were barcoded and pooled in equal concentrations for sequencing. The amplicon pool was purified with the MO BIO UltraClean PCR cleanup kit (https://www.qiagen.com/us/shop/new-products/ultraclean-96-pcr-cleanup-kit/, accessed on 21 September 2021) and sequenced on the Illumina MiSeq sequencing platform 9 (https://www.illumina.com/systems/sequencing-platforms/miseq.html, accessed on 21 September 2021). Via Qiita, sequences were demultiplexed using QIIME 1.9.1 split_libraries using the Golay error-correcting properties of the EMP primers, with default parameters (1.5 max_barcode_erros, 3 max_bad_run_length, 3 phred_quality_threshold, rev_comp_barcode set to true, golay_12 error correction scheme). Sequences were trimmed to 150 bp then further quality processed and denoised using deblur 1.1.0 with default parameters in qiita (mean per nucleotide error rate 0.005, indel probability 0.01, default positive and negative filtering databases). SEPP phylogenetic insertion was used against the Greengenes_13.8 database.

The data generated in this study are available publicly in Qiita under the study ID 11135 (https://qiita.ucsd.edu/study/description/11135, accessed on 21 September 2021), and sequence data associated with this study have been deposited at EBI/ENA under accession number ERP122366.

Statistical Analyses. Group differences in background characteristics (i.e., demographics, neuromedical characteristics) and microbiome diversity were examined using analysis of variance (ANOVA), Wilcoxon/Kruskal–Wallis tests, and Chi-square statistics as appropriate. Pearson’s r was calculated to characterize correlations between microbial diversity and plasma occludin levels. Occludin, sCD14, and LPS values were log_10_ transformed to normalize their distributions. Standard least squares multivariable regression models were constructed to assess potential covariates and interactions. Analyses were conducted using JMP Pro^®^ version 15.0.0 (SAS Institute Inc., Cary, NC, USA, 2018).

## 3. Results

Participants were 50 PWH and 52 PWoH individuals, mean ± SD age 45.6 ± 14.5 years, 28 (27.5%) women, 50 (49.0%) non-white race/ethnicity. Table 1 reports additional details about the samples. Among PWH, 93.9% took combination antiretroviral therapy (ART), 82.0% were virally suppressed, and the median (interquartile range) current and nadir CD4 were 673 (518, 916) and 299 (165, 457), respectively. PWH had significantly higher Faith’s PD (14.2 ± 4.06 vs. 11.7 ± 3.27 *p* = 6.71 × 10^−4^). This principally was driven by MSMs, who comprised 52% of the entire group of PWH (14.4 ± 3.77 vs. 11.3 ± 3.34 in non-MSM; *p* = 3.15 × 10^−5^). Among PWH, MSM and non-MSM did not differ with respect to diversity (for Faith’s PD, 14.4 ± 3.98 versus 13.4 ± 4.76; *p* = 0.544; for Shannon, 4.66 ± 1.05 versus 4.78 ± 1.15; *p* = 0.788). However, among PWoH, diversity as measured by Faith’s PD was higher among MSM than non-MSM (14.6 ± 2.91 versus 11.0 ± 2.99, *p* = 0.0012). This was not the case for Shannon diversity (4.53 ± 0.768 versus 4.85 ± 0.881; *p* = 0.2567). Shannon alpha diversity levels did not differ by sex (females, 4.62 ± 0.886; males, 4.64 ± 0.944, *p* = 0.905). However, Faith’s PD was higher in males than in females (13.5 ± 3.90 vs. 11.4 ± 3.43, *p* = 0.0121). This was entirely due to higher Faith’s PD in MSM (14.4 ± 3.77); non-MSM males did not differ from females (11.3 ± 3.31 vs. 11.4 ± 3.43, *p* = 0.903).

Log_10_ plasma occludin levels were not related to HIV serostatus (PWH, 1.84 ± 0.311 versus 1.85 ± 0.274; *p* = 0.843), age (r = −0.142; *p* = 0.155), sex (females, 1.85 ± 0.0553; males, 1.84 ± 0.034), ethnicity (Black 1.86 ± 0.269; Hispanic 1.94 ± 0.341; Non-Hispanic white 1.78 ± 0.260; other 1.94 ± 0.220; *p* = 0.197), ART (on 1.86 ± 0.299; off 1.61 ± 0.523; *p* = 0.176), estimated duration of HIV infection (r = −0.0706, *p* = 0.634), viral suppression (yes 1.84 ± 0.309; no 1.87 ± 0.356; *p* = 0.812), current CD4 (r = 0.0489; *p* = 0.747), nadir CD4 (r = 0.226; *p* = 0.115), CD4 recovery (r = −0.0999, *p* = 0.509), diabetes mellitus (yes 1.83 ± 0.342; no 1.85 ± 0.289; *p* = 0.852), hypertension (yes 1.87 ± 0.327; no 1.84 ± 0.280; *p* = 0.647), or hyperlipidemia (yes 1.84 ± 0.280; no 1.84 ± 0.305; *p* = 0.660). PWH had numerically, but non-significantly higher levels of plasma sCD14 than PWoH individuals (6.16 ± 0.168 vs. 6.05 ± 0.227; *p* = 0.0548). PWH also had nonsignificantly higher plasma LPS levels than PWoH (1.45 ± 0.325 vs. 1.29 ± 0.350; *p* = 0.106). In PWH, higher plasma occludin correlated with higher LPS (r = 0.427; *p* = 0.0424) and sCD14. PWH with higher levels of plasma sCD14 had higher occludin levels (r = 0.431; *p* = 0.0154). Diversity was not related to viral suppression (yes 13.7 ± 3.36 versus no 16.0 ± 5.85; *p* = 0.122).

In the full cohort, lower gut microbial diversity (Faith’s PD) correlated with higher plasma occludin levels (r = −0.251; *p* = 0.0111). A multivariable model demonstrated an interaction (*p* = 0.0459) such that the correlation held for PWH (r = −0.434; *p* = 0.0017), but not for PWoH individuals (r = −0.0227; *p* = 0.873) (Figure 1). When limiting PWH to those who were virally suppressed, the correlation remained significant (r = −d0.383; *p* = 0.0134). In a multivariable model that controlled for viral suppression in PWH, Faith’s PD was significant (*p* = 8.9 × 10^−4^) while viral suppression was not (*p* = 0.268; model R^2^ = 0.410). Similarly, in multivariable models, sex (*p* = 0.655), age (*p* = 0.183) and ethnicity (*p* = 0.212) were all nonsignificant. To further corroborate the Faith’s PD findings, we evaluated another measure of alpha diversity, the Shannon index. Shannon diversity did not significantly differ between PWH and PWoH (4.68 ± 1.00 versus 4.59 ± 0.792; *p* = 0.6422). In univariable analyses including all participants, the correlation between Shannon’s index and plasma occludin levels did not reach statistical significance (*p* = 0.0653). However, in a multivariable model predicting plasma occludin levels, the Shannon index showed a pattern similar to Faith’s PD: the interaction between Shannon and HIV serostatus demonstrated a trend (*p* = 0.0535; main effect for HIV serostatus, *p* = 0.891; main effect for Shannon, *p* = 0.194), and separate correlations showed that lower Shannon correlated with higher occludin in PWH (r = −0.351, *p* = 0.0126), but not in PWoH (r = 0.0595, *p* = 0.675).

The correlation between Faith’s PD and plasma occludin in PWH was driven by men who have sex with men (MSM). Thus, in a multivariable model with Faith’s PD and MSM as predictors of plasma occludin, the interaction between Faith’s PD and MSM was significant (*p* = 0.0012); among MSM (N = 26), the correlation was r = −0.3834 (*p* = 0.0046), while for non-MSM (N = 13) it was -0.0251 (*p* = 0.864). Only 3 participants were self-described as bisexual.

Table 2 shows the relationship between comorbidities and the key independent and dependent variables for this study (Faith’s PD, Shannon, occludin, sCD14, LPS). Participants with hyperlipidemia had higher Faith’s PD than those without (13.4 ± 3.77 versus 11.5 ± 3.95; *p* = 0.0353); the same was true for those with diabetes mellitus (13.3 ± 3.70 versus 10.0 ± 4.34; *p* = 0.009). No other differences were statistically significant. Among PWH, in multivariable models, the relationship between Faith’s PD and occludin levels remained significant and was not influenced by including hyperlipidemia or diabetes mellitus or their interactions as covariates.

Among PWH taking antiretrovirals, most (70.2%) were on integrase strand inhibitor (INSTI)-based regimens. Other regimen types included protease inhibitor (PI)-based (8.5%), non-nucleoside reverse transcriptase inhibitor (NNRTI)-based (6.4%), NRTI only (4.3%), and other (10.6%). Regimen type was not related to Shannon alpha diversity (*p* = 0.731), to Faith’s PD (*p* = 0.754), or to occludin levels (*p* = 0.843).

Three participants took antibiotics (ampicillin, azithromycin, isoniazid, one each); all were PWH. Those on antibiotics had higher Faith’s PD than those not taking antibiotics (18.2 ± 6.04 versus 12.8 ± 3.73, *p* = 0.0157). There were no statistically significant differences in Shannon diversity according to antibiotic use (5.03 ± 1.25 versus 4.62 ± 0.919, *p* = 0.459). Log_10_ plasma occludin levels were lower in those taking antibiotics (1.51 ± 0.264 versus 1.86 ± 0.287, *p* = 0.0400). In a multivariable model including antibiotic use, Shannon diversity remained significantly associated with occludin levels (*p* = 0.0330), which antibiotic use was not (*p* = 0.126).

## 4. Discussion

We found evidence that lower gut microbial alpha diversity, as indexed by both the Faith’s PD and Shannon indices, was linked to damage to the gut barrier as indexed by higher plasma occludin levels in PWH, but not in PWoH. These findings are consistent with our speculation that a more diverse gut microbiome ensures the presence of the enzymes required to synthesize beneficial metabolites involved in maintaining gut integrity. We suggest that compromise of the gut barrier leads to compensatory overexpression of occludin to facilitate tightening the barrier [28]. Excluding PWH with detectable viral loads or adjusting for viral suppression in a multivariable model did not alter the pattern or significance of this finding. We also found that PWH had significantly higher gut microbial diversity. This relationship was principally driven by MSM, who have been previously demonstrated to have high gut microbial diversity [32,33,34]. Furthermore, in PWH, higher occludin levels were associated with higher levels of plasma sCD14 and LPS, established markers of microbial translocation. These associations were not confounded by demographics or HIV treatment or disease variables.

While higher occludin was related to reduced gut microbial diversity and separately to elevated markers of microbial translocation, the latter two were not significantly correlated with each other. A potential explanation for this is that the microbial translocation markers are at least one step removed from diversity in the hypothesized causal pathway leading from gut microbial diversity to abnormal gut barrier permeability through disturbances in the levels of microbial products that can damage or support the gut barrier. Additionally, it is worth noting that sCD14 is a marker of monocyte activation, which may be influenced by factors other than microbial translocation [35]. Additionally, it is likely that microbial translocation is at least partly dependent upon the specific composition of the gut microbiota, rather than only the level of barrier dysfunction; but, as we previously point out, a comprehensive analysis of alterations in microbial composition was beyond the scope of our hypothesis-driven analyses.

Occludin is an NADH oxidase that influences critical aspects of cell metabolism such as glucose uptake, ATP production, and gene expression [5]. Occludin is present not only in epithelial/endothelial cells, but also is found in large amounts in cells with a very active metabolism, such as pericytes [36] and neurons and astrocytes [37]. Thus, the elevations we saw could be related to changes in these other cell types. However, occludin of epithelial origin would not be expected to affect microbial translocation as indexed by LPS and sCD14 levels, nor would gut microbiome diversity be predicted to affect occludin levels originating from other cell types. Thus, we believe it is most likely that the increased occludin levels seen here reflect altered gut barrier integrity. We are not aware of a way to differentiate sources of plasma occludin, short of doing tissue biopsies and staining. Whether the occludin measured here represents turnover in expressing cells or release due to cell death is unknown.

Our results complement and extend previous work. For example, in one study, caveolar-mediated endocytosis of occludin by Escherichia coli toxin cytotoxic necrotizing factor-1 altered intestinal barrier function in association with downregulation of occludin [38]. In another, application of the proinflammatory cytokine IL-1β at physiologically relevant concentrations caused a significant increase in intestinal TJ permeability [39]. HIV infection leads to early disruption of the gut epithelial barrier and increased intestinal permeability that persists despite viral suppression on ART [15]. Deterioration of the gastrointestinal barrier in PLWH and SIV-infected rhesus macaques has been shown to cause translocation of microbial antigens including LPS, leading to systemic inflammation [17,40]. Additionally, LPS has been shown to decrease TJ protein expression, which is associated with a leaky gut barrier [41].

This study has several limitations. The sample sizes were small, the numbers of women and non-MSM men were low, and although the cohort was diverse, all participants were recruited at a single site, reducing representativeness. We did not measure other markers of microbial translocation and inflammation such as LBP, I-FABP, and sCD163. Although we did not examine the specific microbial taxa associated with occludin levels as this was beyond the scope of our objectives here, the issue deserves careful consideration in future studies. The extent of phylogenetic diversity may have been affected by our use of 16S rRNA amplicon sequencing, rather than metagenomics, which provides improved confidence in enumerating species [42]. However, we have developed a 16S rDNA analysis procedure that does not merge unique 16S reads into OTUs, but rather as the closest named species and percent identity score. This allows a higher resolution of taxonomic assignment than traditional OTU clustering methods [43]. Although diet may influence the microbiome, we did not collect information on dietary differences between the groups. We did not record the time of last meal, so we could not evaluate potential variability in occludin levels related to time of last meal. It is possible that comorbid conditions other than HIV, such as gastrointestinal (GI) malignancies or inflammatory bowel disease, contributed to abnormal gut barrier and hence elevated occludin levels in our study participants. If distributed unevenly between PWH and PWoH, these could bias the study’s findings. However, none of our participants had active GI malignancies and there is no reason to predict that inflammatory bowel diseases would be differentially distributed between PWH and PWoH. We noted that the correlation between Faith’s PD and plasma occludin in PWH was driven by men who have sex with men (MSM). However, as only 7 PWH were non-MSM, our study was not powered to compare the correlation coefficients for the MSM and non-MSM; both were negative correlations (lower PD, higher occludin).

In both diabetes and hyperlipidemia, Faith’s PD values were lower than in those without these disorders. These findings recapitulate previous reports [44,45,46,47,48,49]. Whether these are causal links is not known.

We did not find any associations between plasma occludin levels and markers of HIV disease severity and its successful treatment (current and nadir CD4, viral load). A previous study [50] found that expression of some tight junction proteins (cadherin 1, zonula occludens protein 1, claudin 1, and claudin 7) in gut biopsies was significantly lower in HIV-infected patients with incomplete CD4^+^ recovery. We did not find a correlation between occludin and CD4+ recovery.

Multiple bacterial-viral interactions might play a role in driving the changes to occludin levels and inflammation that we observed here. Such bacterial-viral interactions would likely depend, at least in part, on the presence of viral proteins or whole viral particles. Comparative analyses between the gut microbiomes have shown that the compositions are changed before and after starting ART [51,52]. However, the effects of ART on microbial diversity are difficult to disentangle from those of the viral suppression and immune recovery that ART produces. In the context of our own data, there was no association between viral suppression and gut microbial diversity, although power to detect this was low because only 18% of PWH were not virally suppressed. However, such interactions might still occur, particularly in gut lymphoid tissue which hosts the largest reservoir of latent HIV [53] and is in close proximity to the gut microbiota. In fact, latently infected cells have been shown to express viral proteins at a low level [13,14,54,55], and impaired mucosal epithelial barrier integrity due to persistent gut lymphoid CD4+ T cell depletion [56] would permit microbial translocation and, therefore, promote bacterial-viral interactions. We did not perform gut biopsies to enable assessment of these potential bacterial–viral interactions. Another potential mechanism by which bacterial–viral interactions might occur is through absorption of bacterial metabolites into gut lymphoid tissue. Prior studies have demonstrated that bacterial metabolites can reactivate latent HIV-1 proviruses [57,58] likely by altering the epigenetic status of proviral DNA within the genome of infected cells.

Our results raise the possibility that interventions to ameliorate gut dysbiosis, such as pre- or pro-biotic supplementation or fecal microbial transplantation, might benefit gut barrier integrity and reduce microbial translocation. Since increased microbial translocation is a major driver of inflammation, which is in turn linked to increased mortality and neurocognitive impairment [20,59,60,61,62], such interventions could benefit overall health and neurocognitive function in PWH.

## Figures and Tables

**Figure 1 viruses-13-01891-f001:**
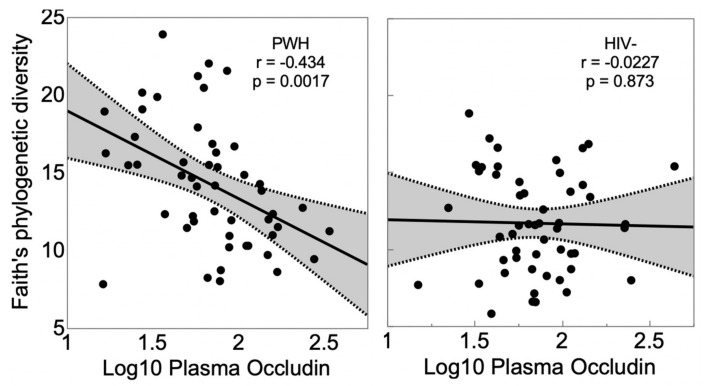
Lower gut microbiome diversity as indexed by Faith’s PD was associated with higher plasma occludin levels in PWH but not in HIV uninfected (PWoH).

**Table 1 viruses-13-01891-t001:** Demographic and clinical characteristics of the study participants.

	PWoH	PWH	*p*
N	52	50	
Age—years (mean ± SD)	46.4 ± 17.2	44.8 ± 11.2	*p* = 0.576
Sex female (N, %)	26 (50.0%)	2 (4.0%)	*p* < 0.0001
Ethnicity Black (N, %)	8 (15.4%)	7 (14.0%)	*p* = 0.206
Hispanic	15 (28.8%)	16 (32.0%)	--
Non-Hispanic White	28 (53.8%)	24 (48.0%)	--
Other	0 (0.0%)	3 (6.0%)	--
Current CD4 (median, IQR)	--	673 (518, 916)	--
Nadir CD4 (median, IQR)	--	299 (165, 457)	--
HIV duration—years (median, IQR)	--	9.71 (4.12, 21.2)	
On antiretroviral therapy (N, %)	--	46 (93.9%)	--
Undetectable plasma viral load in those on ART	--	41 (84.6%)	--
Diabetes mellitus (N, %)	4 (8.6%)	6 (11.5%)	0.570
Hypertension (N, %)	10 (19/2%)	18 (36.7%)	0.0485
Hyperlipidemia (N, %)	9 (17.3%)	14 (28.6%)	0.176
Hepatitis C virus seropositive (N, %)	0	0	1.00

**Table 2 viruses-13-01891-t002:** Levels of key independent and dependent variables according to comorbidity status.

	Hypertension		Hyperlipidemia		Diabetes Mellitus	
	No *n* = 73	Yes *n* = 28	No *n* = 78	Yes *n* = 23	No *n* = 91	Yes *n* = 10
Faith’s PD	12.9 ± 3.69	13.1 ± 4.39	13.4 ± 3.77	11.5 ± 3.95 *	13.3 ± 3.70	10.0 ± 4.34 *
Shannon	4.58 ± 0.91	4.80 ± 0.97	4.72 ± 0.90	4.39 ± 0.004	4.69 ± 0.906	4.16 ± 1.01
log_10_ Occludin	1.84 ± 0.28	1.87 ± 0.33	1.84 ± 0.31	1.87 ± 0.25	1.85 ± 0.289	1.83 ± 0.342
log_10_ sCD14	6.10 ± 0.21	6.16 ± 0.17	6.12 ± 0.21	6.11 ± 0.16	6.11 ± 0.198	6.13 ± 6.38
log_10_ LPS	1.34 ± 0.35	1.45 ± 0.32	1.33 ± 0.34	1.50 ± 0.36	1.35 ± 0.336	1.58 ± 0.418

* *p* < 0.05.

## Data Availability

Data will be made available upon request.

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
