# Peer review of "Markers of Gut Barrier Function and Microbial Translocation Associate with Lower Gut Microbial Diversity in People with HIV"

_viruses, 2021, doi:10.3390/v13101891_

Round 1

Reviewer 1 Report

In the present paper, Ellis et al. assess the link between microbial diversity and gut barrier damage. The authors demonstrate that lower gut microbial alpha diversity (Faith’s PD, Shannon) was linked to GI barrier damage (plasma occludin).

The paper is overall well-written, the methods are appropriate and the results presented deserve publication.

My comments are the following:

1)  The only exclusion criteria appears to be the presence of neurological disorders. Given that the microbiome may vary substantially in subjects with chronic GI diseases (i.e. Crohn's disease) or GI cancer, the authors should specify whether individuals with such conditions were actually included in the study. Similarly, were HIV-infected subjects co-infected with STI (i.e. syphilis, N. gonorrhoeae, C. trachomatis, HPV) and/or received recent antibiotic treatment? This information should also be included in the manuscript.       

2) Which antiretrovirals were used in the patients on cART?

3) Authors should also cite:

  • doi:10.1371/journal.ppat.1004198
  • doi: 10.1097/QAD.0000000000001015 

Author Response

1)  The only exclusion criteria appears to be the presence of neurological disorders. Given that the microbiome may vary substantially in subjects with chronic GI diseases (i.e. Crohn's disease) or GI cancer, the authors should specify whether individuals with such conditions were actually included in the study. Similarly, were HIV-infected subjects co-infected with STI (i.e. syphilis, N. gonorrhoeae, C. trachomatis, HPV) and/or received recent antibiotic treatment? This information should also be included in the manuscript.

We excluded individuals with neurological disorders other than HIV and also those with active psychiatric or substance use disorders (eg, psychosis) that might interfere with completing study evaluations. We collected systematic neuromedical histories to document any comorbid conditions. No participants had active GI cancers. While some may have had chronic GI inflammatory conditions unrelated to HIV, such as Crohn’s disease, there is no reason to suspect that these were differentially distributed between PWH and PWoH. We have added consideration of these issues to the discussion.

We have added the following to the manuscript: Three participants took antibiotics (ampicillin, azithromycin, isoniazid, one each); all were PWH. Those on antibiotics had higher Faith’s PD than those not taking antibiotics (18.2 ± 6.04 versus 12.8 ± 3.73, p =  0.0157). There were no statistically significant differences in Shannon diversity according to antibiotic use (5.03 ± 1.25 versus 4.62 ± 0.919, p = 0.459). Log10 plasma occludin levels were lower in those taking antibiotics (1.51 ± 0.264 versus 1.86 ± 0.287, p = 0.0400). In a multivariable model including antibiotic use, Shannon diversity remained significantly associated with occludin levels (p = 0.0330), which antibiotic use was not (p = 0.126).

2) Which antiretrovirals were used in the patients on cART?

We have added the following to the manuscript: Among PWH taking antiretrovirals, most (70.2%) were on integrase strand inhibitor (INSTI)-based regimens. Other regimen types included protease inhibitor (PI)-based (8.5%), non-nucleoside reverse transcriptase inhibitor (NNRTI)-based (6.4%), NRTI only (4.3%), and other (10.6%). Regimen type was not related to Shannon alpha diversity (p = 0.731), to Faith’s PD (p = 0.754), or to occludin levels (p = 0.843).

3) Authors should also cite: doi:10.1371/journal.ppat.1004198. doi: 10.1097/QAD.0000000000001015

We have incorporated consideration of these two reports to the Introduction and Discussion sections, respectively.

Reviewer 2 Report

The Author, conducted a study with the aim to  evaluate the relationship of gut barrier integrity to gut microbial  diversity in HIV patients.The Author compared the results with HIV negative population. By the results the Author reports that higher occludin levels were associated with higher levels of markers of microbial translocation in HIV patients underling that the damage to the gut epithelial barrier is an important mediator of microbial translocation and inflammation in patients with HIV and that reduced gut microbiome diversity may have an  important role.

My questions are:

1) The percentage of females enrolled in the group of HIV patients are 4% compared to 50% in the control group; this difference can represent a bias in the results of the analysis of the gut microbial diversity, can the Author speculate about that?

2) Can the Author specify if HIV patients with undetectable HIV-RNA at the enrollment were persistently with suppressive viremia after the beginning of the therapy?

3) Can the Author detail the kind of antiretroviral therapy assumed by patients?

4) Are all enrolled patients with diagnosis of chronic infection or are included patients with acute infection? If yes, is the response of patients with acute infection different compared to that with chronic ?

5) Can the Author speculate the result :"...the correlation between Faith’s PD and plasma occludin in PWH was driven by men  who have sex with men (MSM)..." in the discussion?

Author Response

1) The percentage of females enrolled in the group of HIV patients are 4% compared to 50% in the control group; this difference can represent a bias in the results of the analysis of the gut microbial diversity, can the Author speculate about that?

In the original manuscript We reported that log10 plasma occludin levels were not related to sex (females, 1.85 ± 0.0553; males, 1.84 ± 0.034, p = 0.843). We have now added the following: Shannon alpha diversity levels did not differ by sex (females, 4.62 ± 0.886; males, 4.64 ± 0.944, p = 0.905). However, Faith’s PD was higher in males than in females (13.5 ± 3.90 vs 11.4 ± 3.43, p = 0.0121). This was entirely due to higher Faith’s PD in MSM (14.4 ± 3.77); non-MSM males did not differ from females (11.3 ± 3.31 vs 11.4 ± 3.43, p = 0.903). In addition, as we previously reported, in a multivariable model sex (p = 0.655) did not influence the relationship between Faith’s PD and plasma occludin levels.

2) Can the Author specify if HIV patients with undetectable HIV-RNA at the enrollment were persistently with suppressive viremia after the beginning of the therapy?

As we did not provide clinical care for these participants, we did not have direct access to previous viral load measurements. However, as most (72%) had taken more than one antiretroviral regimen, it is reasonable to expect that many would have failed at least one prior regimen and therefore that they had not remained suppressed after initiating their first regimen. In fact, however, there were no differences between those who had and had not received multiple regimens with respect to plasma occludin levels, Faith’s PD or Shannon diversity.

3) Can the Author detail the kind of antiretroviral therapy assumed by patients?

As noted in our response to Reviewer #1, among PWH taking antiretrovirals, most (70.2%) were on integrase strand inhibitor (INSTI)-based regimens. Other regimen types included protease inhibitor (PI)-based (8.5%), non-nucleoside reverse transcriptase inhibitor (NNRTI)-based (6.4%), NRTI only (4.3%), and other (10.6%). Regimen type was not related to Shannon alpha diversity (p = 0.731), to Faith’s PD (p = 0.754), or to occludin levels (p = 0.843).

4) Are all enrolled patients with diagnosis of chronic infection or are included patients with acute infection? If yes, is the response of patients with acute infection different compared to that with chronic ?

All of the patients had chronic HIV infection (> 1 year).

5) Can the Author speculate the result :"...the correlation between Faith’s PD and plasma occludin in PWH was driven by men who have sex with men (MSM)..." in the discussion?

As only 7 PWH were non-MSM, our study was not powered to compare the correlation coefficients for the MSM and non-MSM; both were negative correlations (lower PD, higher occludin). We have added this to the revised manuscript. It seems reasonable to speculate that anal sex might contribute to the higher Faith’s PD of MSM through introduction of microbial species not typically found in the gut.

Reviewer 3 Report

I am delighted to review the manuscript “Microbial translocation and plasma occludin levels associate with lower gut microbial diversity in people with HIV.”  The manuscript presents human data that found changes in serum occludin levels and sCD14 in patients, which correlated with the changes in gut microbial diversity. This is an exciting and essential topic; however, I have few concerns associated with the paper, which are enumerated below.

  1. The manuscript title suggests microbial translocation from the gut, but the data only shows translocation of LPS, a bacterial product. An increase in serum LPS is an indication but not clear evidence of microbial translocation.
  2. If I am not wrong, table 2 suggests that diabetes and altered lipid profile is correlated with the diversity in microbes in the gut. Please discuss these results in discussion.
  3. In Line 51 and 53, please add these references to the rest of the references.
  4. Since authors have the microbial data, please show the diversity data, especially which family of bacteria are over and underrepresented in HIV patients. In such cases, bacteria belonging to Enterobacteriaceae increase; however, it will be an excellent addition to the manuscript.

Author Response

(The authors gave the same response as above.)
